# Expression of Histidine Decarboxylase and Its Roles in Inflammation

**DOI:** 10.3390/ijms20020376

**Published:** 2019-01-16

**Authors:** Noriyasu Hirasawa

**Affiliations:** Graduate School of Pharmaceutical Sciences, Tohoku University, 6-3 Aoba Aramaki, Aoba-ku, Sendai 980-8578, Japan; hirasawa@m.tohoku.ac.jp; Tel.: +81-22-795-6809

**Keywords:** histidine decarboxylase, induced histamine, inflammation

## Abstract

Histamine is a well-known mediator of inflammation that is released from mast cells and basophils. To date, many studies using histamine receptor antagonists have shown that histamine acts through four types of receptors: H1, H2, H3, and H4. Thus, histamine plays more roles in various diseases than had been predicted. However, our knowledge about histamine-producing cells and the molecular mechanisms underlying histamine production at inflammatory sites is still incomplete. The histamine producing enzyme, histidine decarboxylase (HDC), is commonly induced at inflammatory sites during the late and chronic phases of both allergic and non-allergic inflammation. Thus, histamine levels in tissues are maintained at effective concentrations for hours, enabling the regulation of various functions through the production of cytokines/chemokines/growth factors. Understanding the regulation of histamine production will allow the development of a new strategy of using histamine antagonists to treat inflammatory diseases.

## 1. Introduction

Histamine is a well-known mediator of inflammation released from mast cells and basophils. To date, many studies using histamine receptor antagonists have shown that histamine exerts its biological actions through four types of receptors: H1, H2, H3, and H4. In particular, histamine induces allergic reactions and inflammation mainly via H1 and H4 receptors and modulates immune responses mainly via H1, H2, and H4 receptors [1,2]. Recently, H4 antagonists were found to have anti-inflammatory, in addition to their well-known anti-allergic effects, in chronic inflammatory models, such as diabetic nephropathy [3], arthritis [4], and colitis [5]. Thus, it has been recognized that histamine plays more roles in various diseases than had been predicted. However, although the molecular mechanisms of histamine released from mast cells are well understood [6], the mechanisms of histamine production in non-mast cells at inflammatory sites have not been fully clarified. In this review, I focused on the regulation of histamine production and the roles of de novo produced histamine in inflammation.

## 2. Biological Effects of Histamine

### 2.1. Signaling and Distribution of Histamine Receptors

Histamine exerts biological effects through four types of G protein-coupled receptors: H1, H2, H3, and H4 receptors [1,7]. H1 receptors are mainly coupled with Gq/11 and induce the activation of phospholipase C and calcium-mediated responses in various cell types. For example, histamine induces the contraction of smooth muscle cells, the activation of vascular endothelial cells, and the production of prostaglandins and cytokines in various cells including inflammatory cells. H2 receptors are coupled with Gs and induce an increase in cAMP production. They are also expressed in many types of cells including parietal cells of the gastric mucosa, as well as immune and inflammatory cells. The activation of H2 receptors negatively regulates immune and inflammatory cells (see below) and induces gastric acid secretion. H3 receptors and H4 receptors are coupled with Gi and are notably expressed in the central nervous system where they negatively regulate the release of histamine and other neurotransmitters. Recently, it was demonstrated that H3 receptors are also expressed in the peripheral nervous system where they regulate bronchoconstriction, pruritus, and inflammation [8]. The activation of H4 receptors causes a decrease in cAMP and an increase in intracellular calcium concentrations. H4 receptors are expressed in mast cells, eosinophils, T cells, and dendritic cells and play critical roles in immune and inflammatory responses. Recently, H4 antagonists were found to exert anti-inflammatory effects in chronic inflammatory models, such as diabetic nephropathy [3], arthritis [4], and colitis [5], in addition to their anti-allergic effects.

### 2.2. Negative Regulation of Immune and Inflammatory Cell Functions by H2 Receptors In Vitro

The effects of exogenous histamine on immune and inflammatory cells have been analyzed in in vitro systems since the 1970s. The induction of anaphylactic reactions by histamine is mediated mainly by H1 receptors, but the inhibition of inflammation is mainly mediated by H2 receptors. For example, histamine inhibits, via H2 receptors, degranulation in mast cells [9,10] and basophils [11], and cytokine production in mast cells [12]. They also inhibit lymphocyte proliferation [13] and chemotaxis of neutrophils [14] and basophils [15]. Activation and cytokine production by macrophages are also inhibited by histamine [16,17,18]. It polarizes Th2-dominant immune responses via dendritic cells [19,20] and induces IL-12 production, and enhances IL-10 production via H2 receptors [17]. In monocytes, histamine inhibits mixed lymphocyte reactions by suppressing the expression of the intercellular adhesion molecule (ICAM)-1 [18,21], which plays critical roles in cell-cell interactions and the activation of immune and inflammatory cells. The IL-18-induced upregulation of ICAM-1 expression in monocytes is also inhibited by histamine [22]. These effects were exerted via the H2 receptor/cAMP/protein kinase A pathway since the actions of histamine was counteracted by an H2 antagonist and protein kinase A inhibitor. In addition, using H1 and H2 receptor knockout mice and antagonists, it has been demonstrated that histamine activates Th1 cells via H1 receptors but inhibits the function of both Th1 and Th2 cells via H2 receptors [23]. 

Despite these extensive examples of the effects of histamine in vitro, the effects of endogenous histamine have not been well studied. The construction of knockout mice for HDC and each histamine receptor [24,25,26], has enabled the study of the role of endogenous histamine in inflammation and immune responses possible.

## 3. Two Modes of Histamine Release

Histamine is synthesized from histidine by histidine decarboxylase (HDC) [24]. In mast cells and basophils, histamine is produced and stored in granules. The stored histamine is released via degranulation induced by immunological stimulation, such as that by antigens. This type of histamine release is induced rapidly and the histamine levels in surrounding tissues reach a concentration in the order of mM. In contrast, histamine production at inflammatory sites is induced slowly and continuously. In this case, histamine is synthesized de novo in various cells via HDC induction and is released without being stored. Such histamine is called ‘nascent histamine’ or ‘induced histamine’. Importantly, in this case, the levels of histamine in the tissues are limited to the µM range, suggesting that induced and stored histamine play different roles (Figure 1).

Immunological or inflammatory stimulus induced release of stored histamine from mast cells and basophils (stored histamine) and/or production of histamine in inflammatory cells (induced histamine). These modes of histamine release have different characteristics and play different roles.

## 4. Histamine Release from Mast Cells

Mast cells store histamine in their granules and release it mainly via IgE dependent degranulation. Mast cells express Fc receptors for IgE (FcεRI) antibodies on their surface. The crosslinking of IgE antibodies bound to FcεRI with the antigen induces the activation of tyrosine kinases, including the src family tyrosine kinases, lyn and fyn, and the spleen tyrosine kinase Syk, followed by the activation of phospholipase C. Phospholipase C hydrolyses membrane phosphatidylinositol diphosphate, resulting in the production of diacylglycerol and inositol-triphosphates. These secondary messengers activate protein kinase C and induce an increase in the cytosolic calcium concentration, respectively, eliciting degranulation within minutes [6]. Histamine is stored in granules by binding to acidic polysaccharides such as heparin and heparan sulfate found in the granules. The exchange of histamine with Na^2+^ leads to degranulation and histamine release. Importantly, degranulation does not induce damage to the mast cells, and they continue to produce histamine and form granules.

The degranulation of mast cells is induced by not only IgE-mediated responses but also by several non-immunological stimulants, including the activated complements C5a and C3a, called as anaphylatoxin and several neuropeptides such as substance P. Neurokinin 1 receptor is known to be the receptor of substance P on mast cells. In addition, it was recently reported that Mas-related G-protein coupled receptor member X2 (MRGPRX2) is involved in the activation of mast cells by neuropeptides, major basic proteins and several peptidic mast cell activators [27]. Interestingly, it has been suggested that MRGPRX2 plays a role in histamine release in chronic inflammatory diseases [28]. Therefore, the histamine stored in mast cell granules may be involved in various diseases, even in conditions in which IgE serum levels are not elevated. 

## 5. Histamine Production In Vivo and the Functions of Induced Histamine

The induction of HDC expression in non-mast cells has been reported in various experimental rat and mouse inflammatory models and the roles of histamine have been found to be more wide-ranging than expected.

### 5.1. Endotoxin-Induced Inflammation

The intraperitoneal injection of LPS; inflammatory cytokines, such as interleukin (IL)-1 and tumor necrosis factor (TNF)-α [29,30,31]; or staphylococcal enterotoxin A (SEA) [32] in mice, induced HDC expression in various tissues such as the liver, spleen, lung, and bone marrow even in mast cell-deficient W/Wv mice, indicating that HDC was induced in non-mast cells. The type of HDC-expressing cells in the liver of LPS-treated mice remains unknown but these were not circulating granulocytes, monocytes, T cells, mast cells, or phagocytic macrophages [33]. The effects of LPS-induced histamine were examined in a similar experimental model of peritonitis with *Escherichia coli* introduced into the peritoneal cavity via cecal ligation and puncture (CLP), by comparing the inflammatory responses in HDC-deficient mice and wild-type mice. It was found that histamine, at least at the inflammatory site, negatively regulated acute inflammation mediated neutrophils and delayed the elimination of bacteria [34]. In contrast, histamine has been implicated in sepsis-induced major organ failure [35]. In the C57BL/6 mouse CLP model, the roles of histamine in organ injury were examined using histamine receptor and HDC knockout mice. Both H1 and H2 receptors were found to be involved in septic lung and liver injury but only H2 receptors were involved in kidney injury [35].

### 5.2. Dermatitis

In dermatitis models, HDC is induced in epithelial tissues. Daily application of 1–10% anionic surfactants sodium dodecyl sulfate (SDS) or sodium laurate for 4 days on the dorsal skin of mice induced chronic pruritus, which was inhibited by an H1 antagonist [36,37]. The surfactant-induced pruritus was also observed in the mast cell deficient W/Wv mice. Interestingly, the treatment with surfactants increased histamine concentration in the epidermis along with the expression of HDC [36]. The HDC-expressing cells were probably keratinocytes since HDC expression was limited to epithelia tissues and because sodium laurate induced histamine production in a 3D culture of human keratinocytes. Thus, the repeated treatment with surfactants induced HDC expression in epithelial tissues and the produced histamine, at the very least, induced pruritus. 

HDC expression in keratinocytes was also observed in atopic dermatitis patients [38]. In an in vitro system, the stimulation of primary human keratinocytes by thymic stromal lymphopoietin, LPS, house dust mite extracts, and TNF-α induced HDC expression [38]. In this system, the inactivation of histamine by antibodies enhanced the expression of filaggrin, a barrier protein that is a differentiation marker of keratinocytes, suggesting that histamine produced by keratinocytes affects keratinocyte differentiation [38].

### 5.3. Allergic Inflammation 

The quantitative measurement of histamine concentration at inflammatory sites is important but difficult in the usual experimental models. Therefore, we developed an air pouch-type allergic inflammation model in rats. Briefly, 8 mL of air was injected subcutaneously in the dorsum of the sensitized rats and 24 h later, the antigen was injected into the air pouch. In this model, the exudate was easily collected from the air pouch and, therefore, quantitative assessment of inflammatory responses such as the mast cell degranulation, leukocyte infiltration and the production of cytokines are possible. In this model, injection of the antigen into the air pouch induced degranulation of mast cells in tissues and an increase in histamine concentration in the pouch fluid within 30 min. Interestingly, we found that the rate of increase of the exudate histamine levels was biphasic with a rapid increase during the anaphylactic phase followed by a gradual rise during the late phase [39]. 

During the anaphylactic phase, histamine was released via mast cell degranulation since disodium cromoglycate, an inhibitor of mast cell degranulation, reduced the rate of histamine release [40]. In contrast, during the late phase, the increase in exudate histamine concentration occurred up to 24 h after the antigenic challenge, and was accompanied by an increase in HDC activity in inflammatory tissues [39]. HDC mRNA was detected in leukocytes which had infiltrated the pouch fluids as well as in the inflammatory tissues surrounding air pouch. In situ hybridization of HDC mRNA revealed high levels of HDC expression in neutrophils [41]. Interestingly, the effect of dexamethasone on histamine release during these phases was apparently different. While dexamethasone failed to inhibit histamine release during the anaphylaxis phase, it did inhibit histamine production during the late phase in a dose dependent manner [42], suggesting that histamine released during different phases of allergic inflammation were regulated differently.

Local administration of pyrilamine, an H1 antagonist, potently inhibited antigen-induced vascular permeability in the anaphylactic phase [40] but not in the late phase [39]. In contrast, neutrophil accumulation was enhanced by the local administration of an H2 antagonist, cimetidine [39], indicating that in the late phase histamine negatively regulated neutrophil infiltration. Similar results were obtained by using HDC deficient mice, where the antigen challenge did not induce an increase in vascular permeability during the anaphylaxis phase but the infiltration of leucocytes in the late phase was significantly higher than in wild-type mice [43]. The levels of a neutrophil chemokine, macrophage inflammatory protein-2 (MIP-2), and of TNF-α in the pouch fluid of the HDC deficient mice were also significantly higher than in wild-type mice indicating that histamine plays a significant role in the negative regulation of neutrophil infiltration via H2 receptors during allergic inflammation [43].

The repeated challenge of Nc/Nga mice with picryl chloride induced chronic allergic dermatitis with scratching behavior. In this model, a continuous increase in the plasma levels of histamine and HDC mRNA in the inflammatory tissues was also observed [44]. H4 antagonist JNJ7777120 attenuated the scratching behavior and improved dermatitis. Importantly, the effects of JNJ7777120 were augmented by the H1 antagonist olopatadine and the combined treatment showed similar therapeutic efficacy to prednisolone. JNJ7777120 also inhibited the expression of thymus and activation-regulated chemokine (TARC), which recruits Th2 cells, as well as nerve growth factor, suggesting that histamine plays critical roles in chronic allergic dermatitis via the H4 and H1 receptors.

### 5.4. Metal-Induced Inflammation

Nickel is a metal that easily induces allergic responses and inflammation. It binds to proteins making novel antigens and also activates transcription factor hypoxia-induced factors [45,46] by inhibiting its degradation. Because the nickel-induced allergic response is classified mainly as a type-IV allergy and because there is no good experimental model, the roles of histamine in metal inflammation and allergy have not been thoroughly examined. To examine the inflammation induced by metallic medical materials, which include nickel, we established a nickel wire-induced inflammation model [47]. In this model, the implantation of a nickel wire (diameter 0.5 mm, length 5 mm, purity 99.98%) on the dorsum of mice induces the elution of nickel ions and causes severe inflammation. Implantation of the wire increased the expression of HDC mRNA in the inflammatory tissues from 4 h to 72 h post implantation as well as the expression of cytokines, such as MIP-2 and monocyte chemoattractant protein-1 (MCP-1), a chemoattractant for monocytes [47,48,49]. In HDC-deficient mice, the Ni wire-induced an increase in MIP-2 mRNA expression, but not MCP-1 mRNA, was significantly higher than in wild-type mice. MIP-2 expression was also enhanced in histamine H2 receptor knockout mice but not in mast cell deficient W/Wv mice, indicating that the Ni wire induced HDC expression in non-mast cells and that the histamine produced reduced neutrophil infiltration through regulation of MIP-2 expression.

### 5.5. Granulation Tissues

Since the 1960s, it has been proposed that histamine production is increased in granulation tissues [50] which induces pruritus and enhances wound healing. To clarify the roles of histamine in granulation formation, we examined two models: the carrageenan-induced air pouch model in rats [51] and the cotton-thread induced proliferative inflammation model in mice [52]. In the former model, a 2% carrageenan solution was injected into the preformed air pouch of rats. Carrageenan strongly induces the formation of granulation tissues and therefore this model is widely used for to examine proliferative inflammation. Histamine levels in the exudate of carrageenan-induced inflammation were elevated and attained a maximum at 24 h post injection, which closely tracked HDC activity in the inflammatory tissues [51]. The increase in exudate histamine concentration within 30 min of the injection, which was observed in the air pouch type allergic inflammation, was not observed in this model. Treatment with the H2 antagonist cimetidine increased infiltration of neutrophils [51], which is consistent with the allergic air pouch-type inflammation model [39]. In addition, incubation of minced granulation tissues in the presence of histamine induced the production of vascular endothelial growth factor (VEGF) [53]. VEGF expression was also induced by a H2 agonist dimaprit, and histamine induction of VEGF was blocked by H2 antagonists, indicating that it was mediated by H2 receptors. The roles of histamine in angiogenesis and granulation tissue formation were confirmed using a cotton-thread-induced proliferative inflammation model, in which cotton thread (no. 8, 1 cm in length) implantation induced apparent angiogenesis with granulation tissue formation. The implantation of cotton-tread induced HDC expression from one to five days post implantation and expression was also observed in the mast cell-deficient W/Wv mice. During immunochemical analysis of HDC-expressing cells, we found that macrophages in the granulation tissue expressed HDC.

VEGF levels in the granulation tissue three days after the implantation of cotton thread were significantly lower in HDC-deficient mice than in wild-type mice as were angiogenesis and granulation tissue formation. The topical injection of histamine or the H2 agonist dimaprit into the lesion of HDC-deficient mice rescued the defective angiogenesis and granulation tissue formation. In contrast, there was no significant difference in granulation tissue formation and angiogenesis between mast cell-deficient W/Wv mice and control mice, indicating that histamine derived from macrophages plays a significant role in the angiogenesis of the inflammatory granulation tissue.

## 6. Regulation of HDC Expression

### 6.1. Biochemical Features of HDC

HDC from mammals was first purified from the mouse mastocytoma cell line P-815 [54] and fetal rat liver [55]. It is estimated that the molecular weight of HDC is approximately 53–55 kDa and that the active form of HDC is a homo-dimer. The 53–55 kD-type HDC is also dominant in stomach. Cloning mouse [56] and rat [57] HDC cDNA indicated that it is translated as a 74 kDa enzyme and then processed to the 53–55 kDa final form. The 74 kDa HDC is located in the cytosol, and processed after being translocated into the endoplasmic reticulum [58]. The C-terminal region of 74 kD HDC, of approximately 20 kDa, leads to the translocation. This processing of HDC was reported to be regulated in a caspase 9-dependent manner [59]. In addition, 74 kDa HDC is known to be unstable as it has a domain which is rich in proline, glutamic acid, serine, and threonine (PEST domain) [60] and was rapidly degraded via the proteasome pathway [61]. Thus, HDC is regulated by post-translational processing in addition to control of transcription. 

### 6.2. HDC Expression in Mast Cells

HDC expression is regulated by several lineage-specific transcription factors. Yatsunami et al. isolated the genomic DNA of human HDC and found a TATA-like sequence, a GC box, four GATA consensus sequences, four CACC boxes and six leader-binding protein-1 binding motif in the promoter region of the gene [62]. The expression of HDC in mast cells and basophils is regulated transcriptionally in concert with cell differentiation and maturation. The one-week culture of a mastocytoma cell line P815 in the peritoneal cavity of mice caused apparent increase in HDC expression and histamine content in the cultured P815 cells. In this model, GATA2, which plays critical roles in differentiation, is also involved in HDC expression [63]. In contrast, NF-E2 negatively regulated HDC expression [63]. Epigenetic regulation of HDC expression has also been reported. Namely, CpG methylation in the promoter region of HDC, which may vary during mast cell differentiation, affects its expression [64,65,66].

HDC expression in mature mast cells is also regulated in multiple ways including by GATA2 [67], and various mast cell stimulants. The binding of IgE to the high affinity IgE receptor on mast cells [68] and chemical stimulation [69,70]—for example through cyclic AMP or the calcium ionophore A23187 [69] or the combination of 12-*O*-tetradecanoylphorbol 13-acetate and dexamethasone—induce HDC expression. In addition, it was reported that histone deacetylase inhibitor butyrate increased the maturation of granules and increased histamine production in P815 cells [71], indicating that histone deacetylase regulates the expression of HDC. These findings give us important information regarding the signaling pathways for HDC expression in mast cells and the multiple ways in which it is regulated during a type I allergic reaction.

### 6.3. HDC Expression in Non-Mast Cells

In addition to mast cells, HDC expression has been induced in macrophages, neutrophils, lymphocytes, keratinocytes, endothelial cells, and smooth muscle cells (Table 1). HDC is rarely expressed in these cells when unstimulated, but expression is markedly induced by various types of inflammatory stimulants. Because these cells do not have granules like those in mast cells, the histamine produced is released immediately, without being stored. To clarify the mechanism by which inflammatory stimulation induces HDC expression in non-mast cells, we examined the regulation of HDC expression in the mouse macrophage-like cell line RAW264 upon exposure to various inflammatory stimulants. Stimulation of RAW 264 cells with a toll-like receptor 4 stimulant lipopolysaccharide (LPS) or chemical stimulants, such as the calcium-ATPase inhibitor thapsigargin and the protein kinase C activator 12-*O*-tetradecanoylphorbol 13-acetate, increased histamine levels in the conditioned medium. This increase was associated with HDC expression. Specifically, maximum levels of HDC mRNA were reached after 4 h and those of HDC protein were reached after 8 h. Western blot analysis showed that RAW264 cells expressed only the 74 kDa, but not 53 kDa isoform of HDC. Pharmacological analysis found that the expression of HDC mRNA and protein in RAW264 cells was inhibited by the MEK inhibitor U0126 [72,73] and the c-Jun terminal kinase inhibitor SP600125 [72], suggesting that HDC expression was regulated by mitogen-activated protein (MAP) kinases, extracellular signal-regulated kinase (ERK), and c-Jun N-terminal kinase (JNK). The inhibitor of p38 MAP kinase, SB203580 inhibited it but the inhibitory effects was less than U0126 and SP600125. The transcription factor SP1, which binds to GC rich regions, also regulates the expression of HDC [74] because mithramycin, an inhibitor of the SP family, reduced both binding of SP-1 to the GC box in the HDC promoter region and HDC expression [74]. Suzuki et al. also showed that an inhibitor of NF-κB suppressed LPS-induced HDC expression in RAW264.7 cells [62]. An interesting point in the induction of HDC in non-mast cells is that the steroidal anti-inflammatory drug dexamethasone potently inhibited HDC expression and histamine production [75]; whereas, dexamethasone hardly inhibits histamine release via mast cell degranulation. This was confirmed in an in vivo allergic inflammation model (see below) and is an important difference between the degranulation-dependent histamine release in mast cells and degranulation-independent histamine release in non-mast cells.

## 7. Conclusions

Histamine is not only released from mast cells and basophils but is also produced by the induction of HDC in non-mast cells. HDC is commonly induced at inflammatory sites in the late and chronic phases of both allergic and non-allergic inflammation. Thus, histamine levels in tissues are maintained at effective concentrations for hours, enabling the regulation of various actions through the production of cytokine/chemokine/growth factors. As described in this review, histamine exacerbates allergic inflammation via H4 receptors, and angiogenesis and fibrosis via H2 receptors. In such a case, the inhibition of HDC will be a new strategy. In contrast, histamine also exerts inhibitory actions on acute inflammation and immune responses via H2 receptors. Thus, H2 agonists are also expected to be applied to the treatment of various inflammatory diseases. To verify their efficacy, it is important to clarify the source and regulation of histamine released in inflammatory diseases, especially in humans. To date, our knowledge of HDC expression in human inflammatory diseases lags behind our knowledge of the roles histamine plays. Clarifying the regulation of histamine production will promote progress in a new strategy using histidine decarboxylase inhibitor and histamine agonists/antagonists to treat inflammatory diseases.

## Figures and Tables

**Figure 1 ijms-20-00376-f001:**
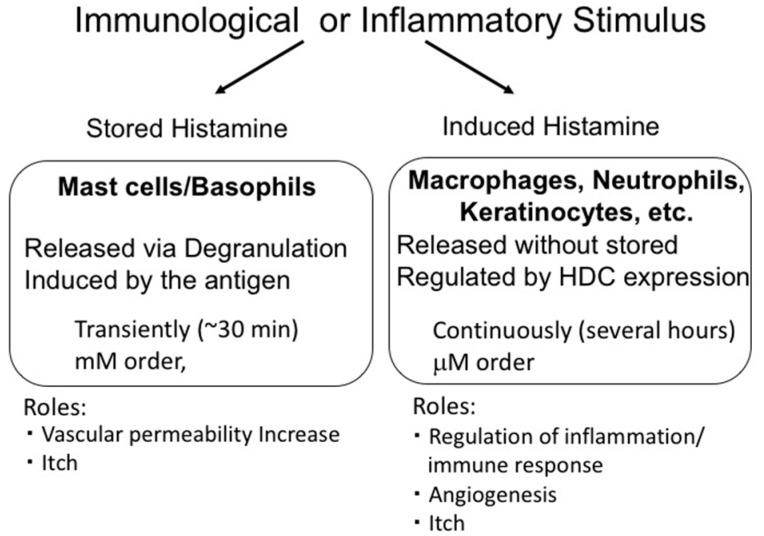
Difference between stored histamine and induced histamine.

**Table 1 ijms-20-00376-t001:** Histidine decarboxylase induction in vitro.

Cells	Stimulants	Ref.
Mast cells	differentiation	[66]
	A23187 + cAMP	[69]
	Dex + TPA	[70]
	granule maturation	[76]
	IgE	[68,77]
Non-mast cells		
Macrophages
peritoneal macrophages		[78]
BMDM	LPS	[79]
RAW264	LPS	[72]
	TPA	[73]
	Thapsigargin	[73]
WEHI-3B	differentiation, LPS	[80]
Neutrophils	LPS	[81]
Lymphocytes		
T cells	Con A	[82]
CD4+, CD8+	Con A	[83]
Jarkat	TPA	[84]
Keratinocytes	SDS	[37]
	LPS, cytokines	[38]
Endothelial cells		[85]
Smooth muscle cells		[86]

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
