# Peer review of "Expression of Histidine Decarboxylase and Its Roles in Inflammation"

_ijms, 2019, doi:10.3390/ijms20020376_

Reviewer 1 Report

This review focuses on histamine and how very little is understood in regards to its regulation in different cell groups.  Histidine decarboxylase (HDC) was supposed to be the focus of this review and its regulation was to be explored.

Major Points:

Overall I recommend reorganizing the review.  I like the introduction; then section 5; I would recommend combining sections 2 and 3 (since they deal with histamine release); then section 6; then section 4.

Figure 1 is confusing.  There is no figure caption-just a title.  I feel that the text under the boxes could be labeled so that the reader could understand what you are trying to reveal (is this the consequences of release?).

Section 4.3  You are speaking of regulation in lines 111-115 could you be more specific.  Are you speaking about protein, RNA or both?

Is HDAC regulated at the level of translation?  If so, you might want to add a few sentences on this.

Expand the section 6.6.  It reads like an after thought.  Maybe add a bit more evidence to show HDC and cancer correlations?

The conclusion is weak.  Is the point to offer HDC as a novel therapeutic target, or just to understand the enzyme better?  It was a bit unclear.

Minor Points:

Lines 28-30:  Sentence is awkward, please reword.

Line 49:  "and" not adb

Table 1 title "Histidine" not histidine

Author Response

Responses:

To Reviewer 1:

Overall I recommend reorganizing the review.  I like the introduction; then section 5; I would recommend combining sections 2 and 3 (since they deal with histamine release); then section 6; then section 4.

Response: Thank you very much for your suggestion.  I reorganized my review according to your suggestion.

Figure 1 is confusing.  There is no figure caption-just a title.  I feel that the text under the boxes could be labeled so that the reader could understand what you are trying to reveal (is this the consequences of release?).

Response: Thank you very much for your suggestion.  I revised the figure and added figure legend.

Section 4.3  You are speaking of regulation in lines 111-115 could you be more specific.  Are you speaking about protein, RNA or both?

Is HDAC regulated at the level of translation?  If so, you might want to add a few sentences on this.

Response: I added an explanation about the effects of HDAC inhibitor and its ref (line 255-257).

Expand the section 6.6.  It reads like an after thought.  Maybe add a bit more evidence to show HDC and cancer correlations?

Response: I deleted this section according to reviewer 2.  I focused this review to inflammation.

The conclusion is weak.  Is the point to offer HDC as a novel therapeutic target, or just to understand the enzyme better?  It was a bit unclear.

Response:  Thank you very much for your suggestion. I revised my conclusion.

Lines 28-30:  Sentence is awkward, please reword.

Response:  Thank you very much for your suggestion.  I deleted the second sentence.

Line 49:  "and" not adb, and Table 1 title "Histidine" not histidine

Response:  Thank you very much for your indication.  I corrected them.

Reviewer 2 Report

The manuscript by Hirasawa is a comprehensive and well writen review of the current knowledge of the role of histamine and it's receptors in inflammation and immune response. I feel the manuscript is apt for publication, given the author addresses the following minor issues:

Paragraph starting on line 39 ("Histamine is synthesized from histidine by histidine decarboxylase (HDC)...." has no references and should have references.

Point 6.6. Tumors is minimal when compared to other sections and should either be removed or further developed.

Author Response

To Reviewer 2:

Paragraph starting on line 39 ("Histamine is synthesized from histidine by histidine decarboxylase (HDC)...." has no references and should have references.

Reponses:  Thank you very much for your suggestion. I added a ref.

Point 6.6. Tumors is minimal when compared to other sections and should either be removed or further developed.

Response:  Thank you very much for your suggestion. I deleted this section.